# Sugary Logistics Gone Wrong: Membrane Trafficking and Congenital Disorders of Glycosylation

**DOI:** 10.3390/ijms21134654

**Published:** 2020-06-30

**Authors:** Peter T. A. Linders, Ella Peters, Martin ter Beest, Dirk J. Lefeber, Geert van den Bogaart

**Affiliations:** 1Tumor Immunology Lab, Radboud University Medical Center, Radboud Institute for Molecular Life Sciences, Geert Grooteplein 28, 6525 GA Nijmegen, The Netherlands; peter.linders@radboudumc.nl (P.T.A.L.); ella.peters@kpnmail.nl (E.P.); martin.terbeest@radboudumc.nl (M.t.B.); 2Department of Neurology, Donders Institute for Brain, Cognition and Behaviour, Radboud University Medical Center, Geert Grooteplein 10, 6525 GA Nijmegen, The Netherlands; 3Department of Laboratory Medicine, Translational Metabolic Laboratory, Radboud University Medical Center, Geert Grooteplein 10, 6525 GA Nijmegen, The Netherlands; 4Department of Molecular Immunology, Groningen Biomolecular Sciences and Biotechnology Institute, University of Groningen, Nijenborgh 7, 9747 AG Groningen, The Netherlands

**Keywords:** post-translational modification, glycosylation, membrane trafficking, Golgi apparatus, secretory pathway, congenital disorders of glycosylation

## Abstract

Glycosylation is an important post-translational modification for both intracellular and secreted proteins. For glycosylation to occur, cargo must be transported after synthesis through the different compartments of the Golgi apparatus where distinct monosaccharides are sequentially bound and trimmed, resulting in increasingly complex branched glycan structures. Of utmost importance for this process is the intraorganellar environment of the Golgi. Each Golgi compartment has a distinct pH, which is maintained by the vacuolar H^+^-ATPase (V-ATPase). Moreover, tethering factors such as Golgins and the conserved oligomeric Golgi (COG) complex, in concert with coatomer (COPI) and soluble N-ethylmaleimide-sensitive factor attachment protein receptor (SNARE)-mediated membrane fusion, efficiently deliver glycosylation enzymes to the right Golgi compartment. Together, these factors maintain intra-Golgi trafficking of proteins involved in glycosylation and thereby enable proper glycosylation. However, pathogenic mutations in these factors can cause defective glycosylation and lead to diseases with a wide variety of symptoms such as liver dysfunction and skin and bone disorders. Collectively, this group of disorders is known as congenital disorders of glycosylation (CDG). Recent technological advances have enabled the robust identification of novel CDGs related to membrane trafficking components. In this review, we highlight differences and similarities between membrane trafficking-related CDGs.

## 1. Introduction

The conjugation of oligosaccharide structures to proteins, glycosylation, is a ubiquitous and fundamental post-translational modification found in all domains of life. Glycosylation is not only important for the structure and function of proteins, but also for their transit and selective targeting through the secretory pathway [1,2,3,4,5,6,7]. In mammals, approximately 700 proteins are necessary for generating the full diversity of over 7000 glycan structures [8,9,10,11,12]. The addition of glycan structures in vertebrates is a sequential process and involves both the addition of monosaccharides via glycosyltransferases and the trimming of glycans by glycosidases [13]. Only ten different monosaccharides are required to build the full glycan spectrum: fucose (Fuc), galactose (Gal), glucose (Glc), N-acetylgalactosamine (GalNAc), N-acetylglucosamine (GlcNAc), glucuronic acid (GlcA), mannose (Man), sialic acid (SA, also known as neuraminic acid), xylose (Xyl), and recently identified ribitol [8,10,11,14].

In vertebrates, N-glycan synthesis is initiated in the ER as a 14 monosaccharide precursor on the carrier lipid dolichol. During translation, this glycan is transferred by oligosaccharyltransferase (OST) [15,16,17] from dolichol to the nascent polypeptide at acceptor peptide sequons, generally consisting of an Asn-X-(Ser/Thr) motif [18,19,20,21]. Distal glucose moieties of these immature, high glucose and mannose containing, glycan structures are subsequently trimmed before Golgi entry; an important step in the control of misfolded glycoproteins in the ER [5,22,23]. Glycoproteins then exit the ER via, for instance, cargo receptor ERGIC-53 [24,25,26] and are transported to the Golgi apparatus for further processing. In the Golgi, glycoproteins are trimmed, extended, and branched until they reach their final glycan form. The mammalian Golgi apparatus is a single large perinuclear organelle, organized into discrete compartments or cisternae [27,28,29]. The Golgi can be subdivided into *cis*-Golgi, closest to the ER, medial-Golgi, *trans*-Golgi, and the *trans*-Golgi network (TGN), furthest away from the nucleus. Furthermore, mammals have a pre-Golgi compartment known as the ER-Golgi intermediate compartment (ERGIC, previously known as the vesicular-tubular cluster (VTC)) [30,31]. Newly synthesized glycoproteins emanating from the ER enter the Golgi apparatus at the *cis*-Golgi, sequentially pass through medial- and *trans*-Golgi, and finally, exit the Golgi at the TGN. The compartmentalization of the Golgi allows for distinct environments containing subsets of glycosylation enzymes [32,33,34,35], enabling sequential modifications for the formation of completely mature glycoproteins. The organization of Golgi-resident enzymes and the Golgi apparatus itself differs between cell types, contributing to glycoprotein diversity [28,36,37,38,39]. Two examples are the distribution of α-mannosidases I and II, which primarily localize to the *trans*-Golgi in intestinal goblet cells, but are distributed over all Golgi cisternae in hepatocytes, the functional consequences of which are currently unknown [39].

Efficient glycosylation fully relies on the correct localization of glycosylation enzymes, as well as on the delivery of other glycosylation machinery, such as nucleotide sugar transporters, and cargo proteins to be glycosylated to the correct Golgi compartment. An important factor involved in the correct trafficking of glycosylation enzymes is the maintenance of pH within the Golgi apparatus. In eukaryotic cells, the principal proton pump for the regulation of intraorganellar pH is the vacuolar H^+^-ATPase (V-ATPase). The membrane V_0_ domain anchors this complex in the membrane, and the V_1_ domain is cytosolic [40]. The V_0_ domain contains six different subunits (a, d, e, c, c’, and c’’). This domain functions as a proton translocator across the membrane, which not only results in a pH gradient, but also in a change in membrane potential, which is neutralized by counter ions such as K^+^ and Cl^−^. The cytosolic V_1_ domain contains eight subunits (A-H), and its main function is ATP hydrolysis [40] to provide the energy needed for the pH gradient. In mammals, the specificity of V-ATPase localization is encoded in the V_0_a subunit, as four unique isoforms exist (V_0_a1-4). This is in contrast to *Saccharomyces cerevisiae*, which has only two unique isoforms (Vph1p and Stv1p) [40,41,42,43,44]. The diversity in V_0_a subunits is likely important for specific cell type-dependent functions and differential regulation of the pH in different organelles. Isoform V_0_a1 is targeted to secretory vesicles and V_0_a2 to the Golgi and endosomes, and V_0_a3, highly expressed in macrophages and osteoclasts [45,46], is enriched in late endosomes and lysosomes, while V_0_a4 is mainly expressed in the kidney, inner ear, and ocular ciliary body [47,48,49,50]. The V-ATPase ensures a constant pH in the various Golgi compartments, which ranges from 6.7 for the *cis*-Golgi to 6.0 for the *trans*-Golgi [51]. Given the pH optima of glycosylation enzymes, this pH gradient could restrict the activity of glycosylation enzymes to their target Golgi compartment [52]. However, this might not be the complete explanation considering the broad distribution of pH optima and the small differences in absolute pH between the cisternae. Instead, or additionally, the pH-sensitive binding and release to cargo adapters might ensure correct enzyme localization to the target Golgi compartment [53,54,55].

Several models for the trafficking routes in the Golgi exist, but the most favorable model of membrane traffic within the Golgi is the cisternal maturation model (Figure 1) [56]. Cisternal maturation is the gradual conversion of a Golgi compartment by the delivery of proteins and lipids from more mature Golgi compartments concomitant with the removal of Golgi proteins and lipids from previous Golgi compartments by coatomer (coat protein complex I; COPI)-mediated retrograde membrane trafficking [27,56]. Before membrane fusion of these COPI vesicles, a set of molecular instruments orchestrates correct vesicle targeting to and within the Golgi. An important group of such trafficking factors is the Golgin family, which consist of large coiled-coil proteins that associate with the Golgi membrane. Golgins form a tentacular web in the cytosol that efficiently and selectively tethers cargo vesicles [57,58,59]. Concurrently, Golgins can act as scaffolding proteins for small Rab or Arf GTPases [60,61,62]. At the Golgi, Rab6 and Rab30 can recruit effectors, such as the cytoskeletal motor protein myosin II, for vesicle trafficking [63,64,65,66,67]. Completing the ensemble is the conserved oligomeric Golgi (COG) tethering complex, a hetero-octameric protein complex bridging the Golgi membrane and COPI vesicles [68,69]. Finally, when the Golgi membrane and the uncoated COPI vesicle are in close enough proximity, membrane fusion occurs. Membrane fusion is performed by soluble N-ethylmaleimide-sensitive factor attachment protein receptor (SNARE) proteins.

As glycosylation is such an extensive process with a multitude of different factors that operate together for sequential remodeling of glycan moieties, only slight disturbances can have major implications on glycosylation. As such, over 100 monogenic diseases have been identified characterized by dysfunctional glycosylation, which form a group collectively known as congenital disorders of glycosylation (CDG) [53,70,71]. A large subset of these includes genetic variants in the prior mentioned trafficking proteins, but also in subunits of the vacuolar H^+^-ATPase and its assembly factors. Recent technological advances in CDG diagnostics have enabled more comprehensive analysis of glycosylation disorders. Novel mass spectrometric methods to detect changes in glycosylation [72,73] together with next-generation sequencing to detect novel genomic mutations [74,75] are a powerful combination for the interrogation of membrane trafficking components in CDGs. This review serves to provide a comprehensive overview of trafficking-related CDGs and to form a detailed understanding of how Golgi trafficking influences glycosylation.

## 2. Membrane Trafficking Components in CDG

Efficient membrane trafficking is of utmost importance for the entire secretory pathway. In this review, we focus on disorders directly affecting Golgi function. Therefore, other disorders affecting for instance ER to ERGIC transport will not be discussed.

### 2.1. Vacuolar H^+^-ATPase

For efficient delivery of glycosylation enzymes to Golgi cisternae to occur, the intraorganellar pH must allow the association of the glycosylation enzyme with a trafficking cargo receptor at the donor compartment and release from the receptor at the receiving compartment. Given the role of pH in intracellular trafficking and how stringent cisternal pH is regulated in the Golgi apparatus (Figure 2) [51], alterations in pH maintenance might be expected to lead to mislocalization of glycosylation enzymes, which in turn cause glycosylation disorders. Indeed, genetic variants in *ATP6V0A2*, the gene encoding the membrane-bound V_0_a2 subunit of the V-ATPase localized to endosomes and TGN [49,50], lead to glycosylation defects [48]. Patients with pathogenic variants, loss-of-function mutations leading to a truncated protein, in *ATP6V0A2* present with wrinkly skin syndrome and autosomal recessive cutis laxa type II, both connective tissue disorders related to the secretion of elastin to the extracellular matrix, and neurological involvement. Moreover, experiments in patient fibroblasts with the fungal metabolite Brefeldin A (BFA), which inhibits the formation of COPI vesicles [76], demonstrate that retrograde intra-Golgi trafficking is impaired in pathogenic *ATP6V0A2* variants, likely causing the mislocalization of glycosyltransferases. Supporting this, neutralization of the Golgi pH with the weak base ammonium chloride or proton pump inhibitor Bafilomycin A1 also mislocalizes glycosyltransferases [77]. Furthermore, the misregulation of Golgi pH could influence the delivery of glycosylated cargo via the cargo receptor ERGIC53, which binds high mannose glycans in the ER in a pH-dependent manner and transports them to the Golgi apparatus [24,25,26,78]. Thus, in addition to the deviation of the pH values away from those optimal for glycosyltransferase activity, mutations in subunits of the V-ATPase could result in mislocalization of both glycosyltransferases and their substrates, and this likely contributes to the CDG pathology. General pH maintenance in the Golgi is imperative for physiological glycosylation.

Aside from variants in the V-ATPase itself, assembly factors and accessory proteins have also been implicated in CDGs. One accessory protein of the V-ATPase, Ac45 (Figure 2; also known as ATP6AP1, the ortholog of yeast Voa1p), is ubiquitously expressed [84,85,86,87] and mainly guides the V-ATPase into cell type-specific subcellular compartments such as neuroendocrine secretory vesicles [88,89] or the ruffled border of osteoclasts [84,90,91]. Missense mutations at sites coding for highly conserved residues in Ac45 have been identified [92] with patients suffering from immunodeficiencies, hepatopathy, neurocognitive abnormalities, and abnormal protein glycosylation. This not only demonstrates that Ac45 is an important factor in (tissue-specific) trafficking of the V-ATPase, but also strengthens the notion that the trafficking of the V-ATPase is imperative for functional protein glycosylation.

Another accessory protein, ATP6AP2 (Figure 2), has also been implicated recently in CDG [93]. While its precise mechanistic role in V-ATPase assembly has not been determined, loss of ATP6AP2 in murine cardiomyocytes, hepatocytes, or podocytes causes a decrease in V_0_a1-3 protein levels, as well as defects in autophagy [94,95,96]. Patients with missense mutations in *ATP6AP2* present protein glycosylation abnormalities with hypogalactosylation and hyposialylation and autophagic defects relating to aberrant lysosomal acidification [93]. This phenotype demonstrates the importance of a functional V-ATPase for both glycosylation and lysosomal function.

Similarly, recently, a novel CDG involving the putative V-ATPase assembly factor VMA21 (ortholog of yeast Vma21p) was discovered, with patients presenting with a hepatic phenotype with steatosis and hypercholesterolemia [97]. Two mutations causing a premature stop codon and one missense mutation in *VMA21* were discovered. Mechanistically, the symptoms of VMA21-CDG patients are the result of impaired lipophagy due to reduced lysosomal acidification, and patients present with a loss of sialic acid and galactose on glycoproteins [97]. In yeast, Vma21p interacts with V_0_ subunit *c’* and thus promotes the assembly of the V_0_ proteolipid subunits into a ring (Figure 2) [79,80]. In mammals, VMA21 can also directly interact with subunit *a* of the V_0_ domain, and this interaction is dependent on glycosylation of subunit *a* (Figure 2) [81,82]. This observation has a two-pronged implication: efficient glycosylation is reliant on the proper assembly of the V-ATPase, and the proper assembly of the V-ATPase is reliant on efficient glycosylation. This could function as a quality control mechanism to abrogate the assembly of a faulty V-ATPase. Most interestingly, in yeast strains lacking Vma21p, the ER-Golgi SNARE Bos1p (ortholog of mammalian GosR2 [98,99]) was completely absent from COPII vesicles [100], strengthening the importance of pH homeostasis for functional ER-Golgi trafficking.

Furthermore, missense mutations in the gene coding for putative V-ATPase assembly factors uncharacterized transmembrane protein 199 (TMEM199, the ortholog of yeast Vma12p) and coiled-coil-domain containing protein 115 (CCDC115, the ortholog of yeast Vma22p) have also been found in novel CDGs [101,102]. CCDC115 and TMEM199 are hypothesized to assemble the membrane-integral V_0_ domain of the V-ATPase through interactions with subunit *a* (Figure 2) [83]. In *S. cerevisiae*, assembly of the V_0_ domain mediated by Vma21p, Vma12p, and Vma22p occurs at the ER membrane [103,104,105], after which the V_0_ domain is transported to the Golgi and associates with the V_1_ domain to form the fully functional V-ATPase complex [40]. In mammals, the exact mechanism of V-ATPase assembly is still unclear, but as TMEM199 and CCDC115 localize to the ER, it is suggested that mammalian V-ATPase assembly is analogous to that in yeast [106]. Pathogenic variants in either protein mainly have a hepatic phenotype similar to VMA21-CDG, although a neurological phenotype is observed for CCDC115-CDG patients. Abnormal glycosylation, mainly the truncation of glycans through the loss of sialylation and galactosylation, is seen for both CDGs. This observation corresponds to the mislocalization of especially *trans*-Golgi-resident galactosyltransferases, which is consistent with the concomitant acidification of the Golgi apparatus. In contrast to VMA21-CDG [97], but in line with ATP6V0A2-CDG [48], no disorders in autophagy were described for TMEM199-CDG and CCDC115-CDG [101,102]. One explanation is that a potential lysosomal defect was not investigated. Another plausible explanation for this phenomenon is that TMEM199 and CCDC115 participate in the assembly of a specific V-ATPase for the Golgi apparatus, while the lysosomal V-ATPase is still assembled regularly in TMEM199-CDG and CCDC115-CDG. This raises the possibility that TMEM199 and CCDC115 primarily interact with a certain V_0_a subunit, likely V_0_a2 concerning the glycosylation phenotypes in TMEM199-CDG and CCDC115-CDG, to facilitate the assembly of a Golgi-specific V-ATPase [107,108,109]. The severity of the symptoms of ATP6AP1-CDG, ATP6AP2-CDG, and VMA21-CDG patients could stem from a more general role of these factors in V-ATPase assembly. Incomplete assembly of the V-ATPase might also result in mislocalization of the V_0_ domain, as the trafficking of the V_0_ domain might be pH dependent itself, thereby amplifying the phenotypes. Aside from their role in CDG, TMEM199 and CCDC115 have also been implicated in iron metabolism [106] and influenza A virus infection, most probably via a pH-dependent mechanism [110]. These observations underpin how V-ATPase assembly, glycosylation, ER-to-Golgi, and intra-Golgi trafficking are intertwined.

Finally, pathogenic variants in the uncharacterized transmembrane protein 165 (TMEM165, also known as TPARL) leading to CDG have been identified [111], one intronic mutation in the gene coding for TMEM165 leading to the production of a truncated protein and two missense mutations at sites coding for highly conserved residues. Patients with TMEM165-CDG mainly present with skeletal and hepatic abnormalities and an N-glycosylation defect consisting of hyposialylation and hypogalactosylation. No O-glycosylation abnormalities were observed in these patients. TMEM165 is a putative proton pump, based on protein sequence homology [111], primarily localizing to the late Golgi apparatus and potentially maintaining the pH in this compartment (Figure 2). This theory is corroborated by the similar glycosylation defects observed with TMEM165-CDG and V-ATPase-related CDGs discussed above. Additionally, depletion of TMEM165 using RNA interference in HEK293 cells revealed a defect in Golgi galactosylation, which could be rescued by supplementation with manganese [112]. As manganese is required for the proper functioning of some glycosylation enzymes [113,114], TMEM165 may also be required for manganese homeostasis, perhaps by functioning as a manganese transporter.

### 2.2. Golgins, GRASPs, GORAB, and Rabs

Vesicle fusion with the Golgi apparatus is initiated by specific capture of vesicles by long coiled-coil proteins decorating the Golgi membrane: Golgins [57,58,59,115]. Next to this, Golgins also act as structural proteins to maintain Golgi architecture [116]. To date, genetic variants in several Golgins have been identified that affect both retrograde intra-Golgi trafficking and glycosylation. For instance, a nonsense mutation in the gene coding for GMAP-210 (Figure 3, also known as TRIP11) causes neonatal lethal skeletal dysplasia in both mice and humans [117]. Similar to GORAB (see below), the phenotypes in patients carrying loss-of-function mutations in *TRIP11* are caused by defective glycosylation of extracellular matrix proteins. GMAP-210 normally functions as a tether for both ER-to-Golgi and intra-Golgi vesicles [118,119]. *TRIP11* mutant chondrocytes and osteoblasts isolated from mice carrying this nonsense mutation showed swollen ER and a disrupted Golgi architecture. Patient fibroblasts with either a heterozygous or homozygous nonsense variant of GMAP-210 showed incomplete glycosylation of the model secretory protein vesicular stomatitis virus G protein (VSVG) fused to GFP, suggesting a function of GMAP-210 in trafficking. These mutant cells also showed increased lectin GS-II binding along cell surfaces, indicating a defect in glycosylation due to GMAP-210 loss-of-function, as lectin GS-II binds to terminal non-reducing N-acetyl-d-glucosamine, which is normally not present in fully processed glycoproteins [117]. Hypomorphic mutations, or partial loss-of-function mutations, in *TRIP11* cause a different genetic disorder called odontochondrodysplasia (ODCD) [120], affecting skeletal and dental development. In contrast to the loss-of-function mutations in *TRIP11*, secretion is not affected in ODCD. Glycans on the lysosomal glycoprotein LAMP2 and the extracellular matrix protein decorin were both abnormal, and synthesis of extracellular matrix proteins was strongly reduced, leading to disease [120].

Other Golgins also regulate glycosylation, but have not been associated with CDGs. An intronic splice donor site mutation in the gene coding for Golgin giantin (Figure 3, also known as GOLGB1) produces a truncated protein, which causes cleft palate in mice, with murine embryos showing an increase in GS-II lectin binding to terminal GlcNAc moieties within the palatal regions, which is indicative of incomplete protein glycosylation. In parallel, frontal sections of developing palatal shelves of giantin loss-of-function mutant mice show increased binding of PNA lectin after desialylation with neuraminidase, showing an increase in galactosylated O-type mucins on the cell surface [121]. In addition to giantin, certain membrane tethering proteins, such as giantin, GRASP55, and GRASP65 (Figure 3), are thought to regulate the rate of retrograde trafficking, likely to assure efficient recycling of glycosylation enzymes to their target Golgi compartment. RNA interference-mediated depletion of giantin in HeLa cells revealed that this causes aberrant fusion of Golgi cisternae [122]. This, in turn, caused a two-fold increase in the mobility of the glycosylation enzyme ManII as measured by fluorescence recovery after photobleaching and accelerated the trafficking of VSVG to the plasma membrane [123]. Furthermore, siRNA depletion of giantin caused overexpression of sialylated glycoproteins at the cell surface [123]. Moreover, depletion by RNA interference of GRASP55/65 accelerated the anterograde trafficking of VSVG, independent of ER stress and unconventional protein secretion, and concurrently decreased the complete glycosylation of VSVG [124].

More recently, several missense and nonsense mutations in *GORAB* have been identified in patients (Figure 3). Although GORAB was previously described to be a Golgin, it is now understood that it is a COPI vesicle coat scaffolding protein that likely engages in vesicle formation and has been associated with the development of the skin and bone disorder gerodermia osteodysplastica (GO) [125,126,127,128]. GO presents with osteoporosis and has a similar elastin deficiency as cutis laxa [129,130]. Importantly, patients have deficient glycosylation of proteoglycans including decorin and biglycan, leading to their pathologies. GORAB is primarily functional at late stages of intra-Golgi trafficking. GORAB promotes COPI recruitment to the *trans*-Golgi through the formation of stable membrane domains. GORAB also scaffolds the catalytically inactive protein kinase Scyl1 [125]. Scyl1 localizes to the ERGIC and *cis*-Golgi budding sites and binds to COPI coats using a C-terminal RKLD sequence, similar to the KKXX COPI-binding motif present in ER transmembrane proteins. The depletion of Scyl1 in HeLa cells by RNA interference disrupts COPI-mediated retrograde trafficking of the KDEL receptor towards the ER [131]. Several missense and nonsense mutations in *SCYL1* have been described, but, in contrast to GORAB, primarily hepatological and neurological phenotypes were observed [132,133,134]. Cellular investigations of patient fibroblasts show an enlarged Golgi morphology as shown by immunofluorescence and impaired retrograde trafficking when perturbed with BFA [132]. Most interestingly, one patient demonstrated hyposialylation of both transferrin and apolipoprotein CIII as measured by isoelectric focusing (IEF) during a liver crisis, but this returned to normal after the crisis had passed [132]. An explanation for this might be that the secretory burden of glycoproteins in this patient was too high during the crisis, leading to abnormal glycosylation. Pathogenic variants in GORAB either affect the binding affinity of GORAB to Scyl1 or affect the assembly of GORAB in membrane domains, leading to the dysfunction of GORAB. Loss-of-function mutations in *GORAB* inhibit the retrieval of *trans*-Golgi resident enzyme ST6GAL1 in a COPI-mediated manner, demonstrating the necessity of GORAB for COPI-mediated intra-Golgi trafficking. Concurrently, *GORAB* mutant fibroblasts from GO patients show a reduced abundance of complex terminally sialylated glycans, suggesting deficient glycosylation by dysfunctional GORAB [125]. Taken together, the observed phenotypes suggest that *GORAB* mutations can be considered CDGs.

Once a vesicle has been captured by a Golgin, effectors such as Rab GTPases are recruited (Figure 3). Rabs exists in a GDP-bound inactive state and are activated by the exchange for GTP through guanine nucleotide exchange factors (GEFs). This nucleotide exchange results in a conformation switch, which enables Rabs to recruit specific effectors required for vesicular trafficking. Genetic variants within Rab proteins are associated with several neurological and metabolic disorders, including Parkinson’s disease and Neumann–Pick’s disease [135]. Certain Rabs play an active role in retrograde Golgi trafficking, such as Rab6, which is required for bidirectional transport of cargo at the Golgi [64]. Furthermore, Rab2 can influence the function of GMAP-210 in COPI vesicle tethering [119], while Rab1, Rab2, Rab4, and Rab6 can interact with members of the COG complex and thereby tether COPI vesicles to the Golgi membrane [136]. Rab1, in particular, has also been identified to regulate the Golgi architecture and function and therefore has an indispensable role in glycosylation [135]. While genetic variants in Rabs or their regulatory proteins have not been implicated in CDGs thus far, it is conceivable that Rab dysfunction alters the identity of COPI vesicles, which results in the mislocalization of glycosylation enzymes. Thus, considering the potential effects of Rab dysfunction on glycosylation, it stands to reason that Rab CDGs are yet to be identified.

### 2.3. Conserved Oligomeric Golgi Tethering Complex

After Golgins mediate the capture of Golgi-destined vesicles from the cytosol, the COG complex functions as a tether to anchor COPI vesicles to the Golgi membrane [137,138,139]. COG subunits have been described to interact directly with COPI coat components, as well as with SNAREs and Rab GTPases involved in Golgi trafficking [68,140,141,142,143,144,145]. Despite the lack of transmembrane domains in COG subunits, they are membrane-associated proteins. The COG complex is a hetero-octameric protein complex, consisting of eight unique subunits COG1-8 (Figure 4). These subunits organize themselves into two distinct lobes: lobe A consisting of COG1-4 and lobe B consisting of COG5-8. This in turn also dictates their localization: lobe A is primarily present on the Golgi membrane, while lobe B localizes to COPI vesicles [145]. The two lobes are bridged by an interaction between COG1 and COG8 through the formation of alpha-helical bundles [68,140]. The COG complex is therefore required for the trafficking of glycosyltransferases and cargo proteins.

Cellular models of COG-subunit deficiencies show an alteration in glycosylation homeostasis [146]. At a cellular level, the depletion of COG subunits 2, 3, 4, 6, 7, and 8 causes the mislocalization of glycosylation enzymes MAN2A1, MGAT1, B4GALT1, and ST6GAL1 to COG complex-dependent vesicles [147,148], demonstrating their necessity in functional glycosylation [149]. Studies in HeLa cells depleted of COG4 by RNA interference also demonstrated glycosylation defects and the mislocalization of vesicles containing COG-interacting proteins (GEARs) around the Golgi [138,140,148]. Of note, the permanent membrane targeting of COG subunits 4, 7, and 8 by fusing them to the transmembrane protein TMEM115 [150] disrupts the ribbon structure of the Golgi as visualized by electron microscopy, causing the swelling of cisternae and an increase of spherical, non-cisternal elements [151]. Nevertheless, membrane-anchoring of COG4 and COG7, but not COG8, rescued the glycosylation defects observed in their respective CRISPR/Cas9 knockout cell models. The N-terminal attachment of membrane-anchored COG8 interfered with overall COG structure and function, impeding the rescue of the observed glycosylation defects [151]. The permanent Golgi anchoring of COG4 and 7 maintains the polarization of *cis*- and *trans*-Golgi markers, but fails to restore a highly organized Golgi structure in COG4 and 7 knockout cells. These data demonstrate that the membrane association of most COG subunits is imperative for their function, and likely assists in the correct retrieval of glycosyltransferases to earlier Golgi cisternae.

Genetic variants causing CDGs have been identified for all COG subunits, except for COG3 [140,149,152]. Most genetic variants in COG subunits are observed in lobe A subunits COG1 and COG4 and in lobe B subunit COG8 (Figure 4). While in most CDGs, N-glycosylation is mostly affected, COG-CDGs show a broader phenotype with defects in both N- and O-glycan biosynthesis [149]. Clinically, patients of COG-CDGs present with prominent incomplete galactosylation and sialylation [149,153], and this is true for variants in both lobe A and B subunits. While the variants mostly consist of missense mutations or truncations for lobe A subunits and full loss-of-function mutations in lobe B subunits, glycan profiles of patients show the same hypogalactosylation and hyposialylation for both types of mutations [154,155]. This raises the question of whether mutations in one COG subunit affect the entire COG complex. Indeed, a variant in one COG subunit is associated with the instability of another subunit within that lobe, leading to the decrease in protein expression of subunits in the same lobe [156]. This has been shown for COG1 [153] and COG2 [152] in lobe A and COG6, COG7 [157,158], and COG8 [159] for lobe B. Moreover, the mutations in *COG1* are associated with a decrease in protein levels of COG8 [153]. Oppositely, a truncation mutant of COG8 also decreases the protein levels of COG1 [160]. The finding that all eight subunits are required for complete COG function is a possible explanation for the similarity in glycan profiles of COG-CDGs (see below). However, COG1- and COG4-CDG phenotypes are relatively mild, while COG7- and COG8-CDG are much more severe. This could indicate a differential necessity of COG lobe B over lobe A.

COG-CDGs have mainly been implicated through genetic variants of COG1 and COG4 for lobe A. Patients with an 80 residue C-terminal truncation of COG1 present with a reduction of galactose and sialic acid moieties on N- and O-glycans. Consistent with this, fibroblasts isolated from these patients demonstrate a reduction of sialic acid incorporation on mucin-type O-glycans [153]. A different variant, an intronic mutation that results in a frameshift and a premature stop codon in exon 7, in *COG1* causes skeletal defects and mental retardation, together with hypogalactosylation and hyposialylation [161]. Moreover, the Brefeldin A assay performed on these patient fibroblasts revealed a retrograde Golgi trafficking defect. In CHO cells, depletion of COG1 by RNA interference leads to both deficient N- and O-glycosylation [138,140]. Fibroblasts from a patient with two heterozygous missense and nonsense mutations, respectively, in *COG4* also showed hypogalactosylation and hyposialylation [162,163]. Effects of these variants on other COG subunits were not investigated. Moreover, a patient with a different missense mutation in *COG4* presented with a similar glycosylation defect, but the instability of the other subunits in COG lobe A was also observed, likely due to the inability to form COG subcomplexes [164]. Lastly, another amino acid substitution in COG4 was identified causing a rare form of primordial dwarfism, but patients notably have normal glycosylation of serum proteins [165]. Instead, the extracellular matrix protein decorin is abnormally glycated in these patients, underlying their pathology.

To date, only a single mutation, an intronic mutation leading to a decrease in *COG5* expression, has been identified in the gene coding for COG lobe B subunit COG5 with mostly neurological symptoms and similar hypogalactosylation and hyposialylation like in other COG defects [154,166,167,168]. Several studies have shown that a single missense mutation in *COG6* leads to severe neurological and hepatological symptoms and can lead to infant mortality [169,170,171,172]. The relative severity of COG6-CDG is likely due to the observed instability of COG lobe B, as COG6-CDG patients also have lower protein levels of COG5 and COG7. Furthermore, decreased levels of the *trans*-Golgi SNARE syntaxin-6 were detected, suggesting that stabilization of syntaxin-6 via the COG complex is necessary for its function [171]. Intronic mutations affecting the mRNA splicing and ultimately causing the decrease of protein levels of COG7 cause similar glycosylation defects, with patients presenting hyposialylation in their N- and O-linked glycan biosynthesis [157,158,173]. Another study observed that one specific variant in *COG7*, which is associated with infantile mortality [174], causes a loss of sialylation on serum transferrin and on cell surface proteins of patient fibroblasts [175]. Moreover, this variant in *COG7* affects Golgi trafficking as shown by the impaired trafficking of ST3GAL1 from the ER to the *trans*-Golgi in patient fibroblasts. A homozygous nonsense mutation in *COG8* results in the formation of a truncated COG8 subunit, lacking 76 C-terminal residues, affecting the interaction between COG8 and COG1. Patients with this variant present with mild hyposialylation [160]. Similarly, a patient with a different genetic variant resulting in the truncation of COG8 with the loss of 47 C-terminal residues showed a similar deficiency in sialylation [159]. COG8-CDG patients present with a severe neurological phenotype and display a fragmented Golgi apparatus in patient fibroblasts. Overall, the COG defects demonstrate the importance of efficient tethering of intra-Golgi vesicles and show that even small changes are enough to destabilize the COG complex, thereby severely affecting glycosylation.

### 2.4. SNAREs

The final step in the delivery of glycosylation enzymes to the Golgi is SNARE-mediated membrane fusion. SNARE proteins are classified by the central residue in their SNARE motif: R-SNAREs have a central arginine residue, while Qa-, Qb-, Qbc-, and Qc-SNAREs have a central glutamine residue. A tight alpha-helical coiled-coil bundle is formed by one of each type of SNARE motif, provided by three or four cognate SNARE proteins. These are present on both the vesicular (e.g., COPI vesicle) and target membranes (e.g., *cis*-Golgi), and the coiled-coil formation provides enough energy to fuse the two opposing membranes [176,177]. At the mammalian ER-Golgi interface, four distinct SNARE complexes exist: Stx5/GosR2 (also known as GS27 or membrin)/Bet1/Sec22b (also known as ERS24) for anterograde transport from ER to ERGIC, Stx5/GosR1/Bet1/Ykt6 for anterograde transport from ERGIC to *cis*-Golgi, Stx5/GosR1 (also known as GS28)/Bet1L (also known as GS15)/Ykt6 for retrograde intra-Golgi transport, and Stx18/Sec20/Use1/Sec22b for retrograde transport from *cis*-Golgi to ER [98,99,178,179,180,181,182,183,184,185,186,187,188,189,190,191]. While in general, R-SNAREs are present on the vesicular membrane in eukaryotic cells [177], evidence suggests that the Qc-SNAREs Bet1 and Bet1L function as the vesicular SNAREs at the ER-Golgi interface [99,188,191,192,193,194]. This observation, together with the necessity of Sec1/Munc18 SM) protein Scfd1 in ER-Golgi SNARE fusion, likely implies specificity to the fusogenic SNARE complex, by inhibiting the formation of non-functional SNARE complexes [30,195,196,197,198].

Despite the essential role of SNAREs in Golgi trafficking [99], there is only limited evidence for a clinical link between glycosylation and genetic variants in SNARE proteins. This also raises the question of whether SNARE CDGs mostly remain undetected due to a possible severity of the disease. Contrary to plasma membrane-localized SNARE complexes, ER-Golgi-localized SNARE complexes lack redundancy, and the loss of a single SNARE protein at this interface might result in detrimental effects for glycosylation and/or life [98]. Recently, the first CDG related to a SNARE protein was identified, namely a point mutation in *STX5*, the gene coding for the Qa-SNARE syntaxin-5 [191]. In animals, the *STX5* gene is transcribed to one mRNA, which produces two different isoforms of Stx5 via an alternative starting codon: 39.6 kDa sized Stx5 Long (Stx5L) and 34.1 kDa sized Stx5 Short (Stx5S) [99,199]. Stx5L is characterized by a 54 residue N-terminal extension with an RKR (arginine-lysine-arginine) ER-retrieval motif and localizes at ER, ERGIC, and *cis*-Golgi. In contrast, Stx5S lacks this RKR ER-retrieval motif and primarily localizes to the Golgi (Figure 4). The point mutation causes the specific loss of Stx5S through the mutation of the second starting methionine residue into valine. This loss of the intra-Golgi dominant Qa-SNARE Stx5S results in a severe disorder, characterized by metabolic and developmental defects and infantile mortality. Microscopy revealed that this is caused by the mislocalization of glycosylation enzymes to the wrong compartment in the Golgi apparatus, leading to the reduced incorporation of galactose and sialic acid moieties in N-glycans and an overrepresentation of immature high-mannose glycans [191].

The Qbc-SNARE SNAP29 has previously also been ascribed a role in Golgi morphology, and its dysfunction could likely result in glycosylation defects [200]. Of note, several missense and truncating mutations in *SNAP29* have been associated with cerebral dysgenesis, neuropathy, ichthyosis, and keratoderma syndrome (CEDNIK, a disorder of brain development, facial dysmorphism, and skin), and Pelizaeus–Merzbacher-like disorder (PMLD, a disorder of brain development and muscle function) [201,202,203,204,205,206,207]. The first report of *SNAP29* involvement in CEDNIK noted that no N- and O-glycosylation defects were observed, although the authors did not describe how the patients were screened [206]. This result does not exclude that SNAP29 is involved in glycosylation, as, assuming that custom IEF methods were applied for CDG screening, it is possible that although glycosylation of either transferrin or apolipoprotein CIII were not affected, other glycosylated proteins were. Additionally, these methods might not have been sensitive enough to detect an underlying glycosylation disorder, or only glycosylation in certain tissues could have been affected. Finally, fusion-impaired forms of GosR2 result in a neurological phenotype in patients with progressive myoclonus epilepsy, but have not been associated with glycosylation defects (Figure 4) [208,209,210]. It is possible that during these studies, no diagnostics for glycosylation has been performed or that through a compensatory mechanism, sufficient glycosylation was maintained. Along these lines, more comprehensive diagnostic screening procedures might implicate more SNAREs in CDGs.

## 3. Discussion and Conclusions

The glycosylation process is an essential part of the secretory pathway and is a complex logistic system. Newly synthesized glycoproteins are shuttled from the ER to the Golgi apparatus, which acts as a production line that sequentially builds complex branched glycan structures on the proteins. Moreover, the Golgi can be considered a distribution center that ensures that glycoproteins are modified and sorted correctly to their final destinations. As with factories and other logistic systems, having efficient infrastructure is the key to a high efficiency and fidelity of production and delivery. Each organelle in the logistic chain needs to function optimally to avoid bottlenecks, and therefore, efficient coordination amongst organelles is of utmost importance. Here, we discussed how dysfunctional transport processes affect glycosylation and, in turn, cause a wide array of symptoms including skin and bone disorders, impaired liver function, and even infantile mortality. Mechanistically, these pathologies are the result of impaired pH homeostasis in the Golgi, incorrect tethering of Golgi-destined vesicles, and defective membrane fusion at the Golgi apparatus (Table 1).

In this review, we discussed pathogenic variants in the V-ATPase and associated proteins that affect Golgi pH homeostasis and glycosylation. Considering that glycosylation enzymes have a broad pH optimum and that the Golgi has a narrow range in pH (pH 6.7 for *cis* to pH 6.0 for *trans*), it seems unlikely that small defects in pH homeostasis result in a pronounced loss of catalytic activity of the glycosylation enzymes. Instead, the prevailing theory is that pH affects the trafficking of glycosylation enzymes to their cognate Golgi compartments and that altering the Golgi pH results in mislocalization of these enzymes to the wrong compartment [53,54,55]. This is corroborated by the observation that Golgi-to-ER recycling mediated by the KDEL receptor is indeed pH-dependent [211] and by the observation that the ER-Golgi SNARE protein Bos1p is absent from COPII vesicles of yeast strains lacking the V-ATPase assembly factor Vma21p [100]. Moreover, it has been described that the luminal pH affects the oligomerization of certain glycosylation enzymes and thereby also influences their localization [212,213,214,215,216,217,218,219]. Another well-characterized example of pH-mediated protein sorting is the binding of the mannose-6-phosphate receptor to the mannose-6-phosphate-labeled cargo destined for the lysosome. Proteins are bound by this receptor in the Golgi and subsequently released in the more acidic environment of late endosomes [220,221,222]. Finally, the pH of the TGN is also important for the correct sorting of the extracellular matrix components laminin and heparan sulfate proteoglycan to the basolateral surface of polarized epithelial cells [220,223,224]. Taken together, evidence shows that homeostasis of pH is of crucial importance for trafficking within, from, and to the Golgi apparatus, and thereby likely a critical factor for glycosylation. This also implicates that mislocalization of the V-ATPase and associated proteins due to pathogenic mutations in trafficking proteins such as SNAREs, COG subunits, or Golgins can affect glycosylation.

Interestingly, except for Stx5-CDG, all described CDGs in trafficking proteins primarily demonstrate defects in late glycan modifications such as galactosylation and sialylation (Table 1). Only in Stx5-CDG, the accumulation of an early-stage high mannose glycan is observed. Why could defects in early glycosylation steps be underrepresented? We can consider several hypotheses. First, correct glycosylation is of absolute importance for development [2,5], and despite the redundancy in glycosylation enzymes, defects affecting early glycosylation reactions might not be conducive to further development. Therefore, potential pathogenic mutations might remain undetected as they are non-viable. Second, as disorders of glycosylation are mostly a secondary effect of trafficking defects, glycosylation disorders might go undiagnosed as they simply have not been screened for in potential cases such as for *GOSR2* mutations [209,210]. Last, early glycosylation defects might go undiagnosed because common biochemical techniques to measure glycosylation in a clinical setting, such as isoelectric focusing of transferrin and apolipoprotein CIII [227], mostly interrogate late glycosylation reactions such as sialylation. Modern advances in clinical diagnostics have added mass spectrometry of intact transferrin [72] and glycomics on total plasma N-glycans [155]. These two approaches have enabled measuring all N-glycan structures on circulating serum proteins, thereby giving a more complete view of the glycosylation defects. This allows clinicians and scientists to identify more complex glycosylation disorders, including those in trafficking factors. Future advances to glycosylation screening might include measuring the glycome of all different tissues in the human body or measuring site-specific glycosylation on a large array of proteins. Understanding trafficking and glycosylation in different cellular contexts will ultimately further our knowledge of glycobiology and eventually increase therapeutic options for patients suffering from CDGs. A subcellular understanding of CDG can pave the road for targeted therapies through, for instance, gene replacement of affected genes or the development of small molecules to modulate ER-to-Golgi and intra-Golgi transport.

## Figures and Tables

**Figure 1 ijms-21-04654-f001:**
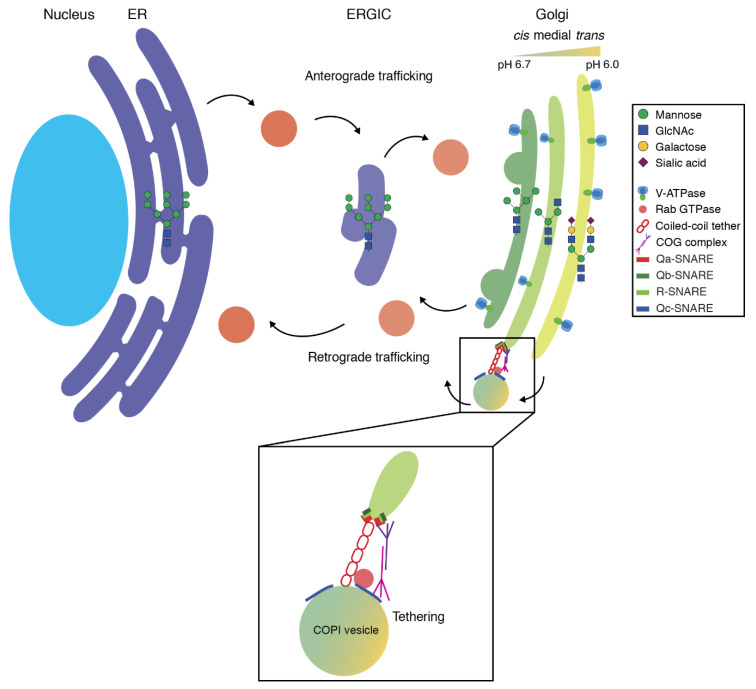
Schematic overview of the early secretory pathway in mammalian cells. Abbreviations: ER, endoplasmic reticulum; ERGIC, endoplasmic reticulum-Golgi intermediate compartment; COG complex, conserved oligomeric Golgi complex; COPI, coat protein complex I.

**Figure 2 ijms-21-04654-f002:**
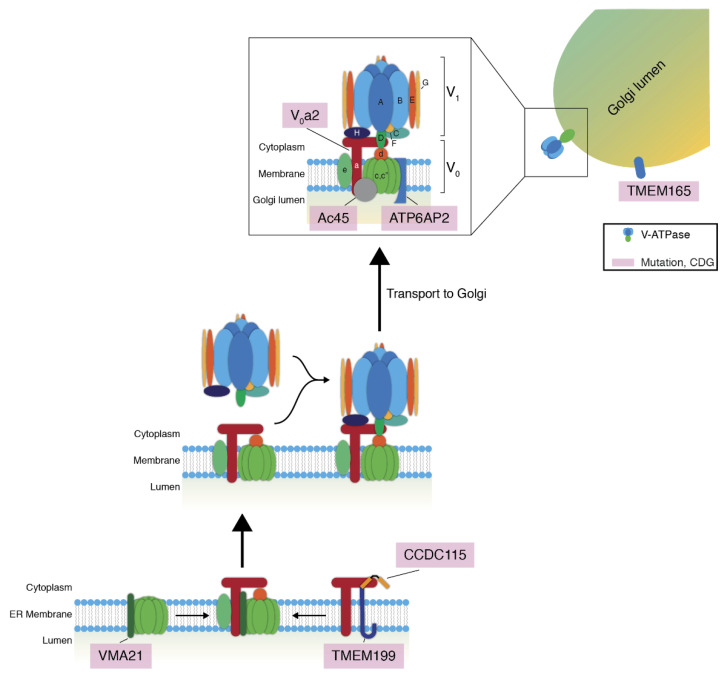
Schematic overview of the mammalian V-ATPase and the putative model of its assembly. Lowercase letters denote the various subunits of the V_0_-domain; uppercase letters denote the subunits of the V_1_-domain. The assembly factors VMA21, TMEM199, and CCDC115 might assemble the membrane-associated V_0_-domain of the V-ATPase. VMA21 interacts with V_0_c’ and V_0_a [79,80,81,82], TMEM199 and CCDC115 interact with V_0_a [83]. Abbreviations: V-ATPase, vacuolar H^+^-ATPase; CDG, congenital disorder of glycosylation.

**Figure 3 ijms-21-04654-f003:**
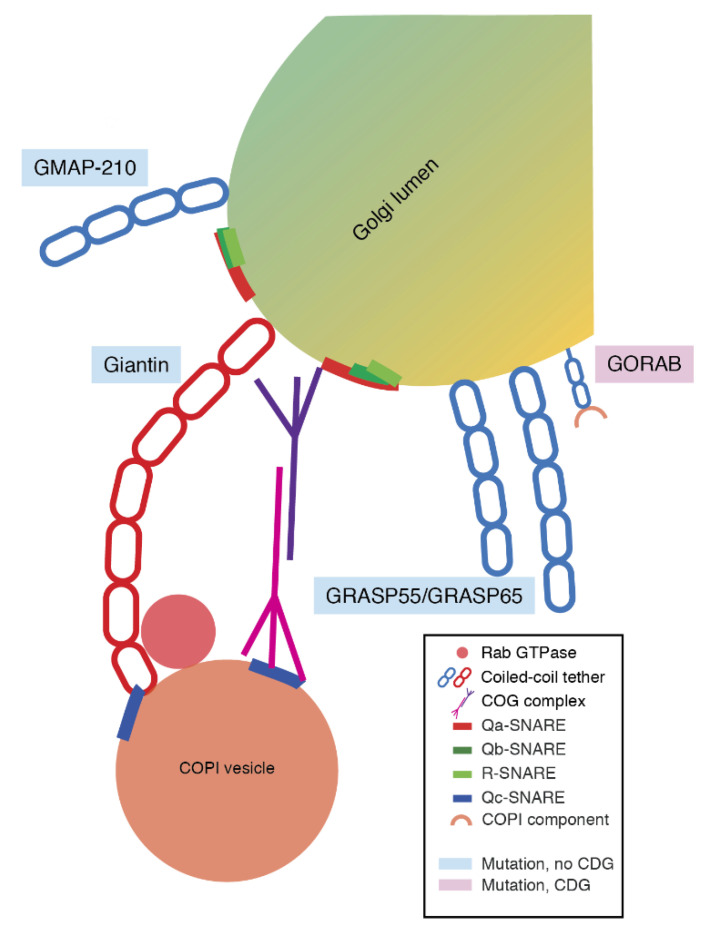
Schematic overview of COPI vesicle capture by coiled-coil tethering proteins at the Golgi. Tentacular coiled-coil proteins attached to the Golgi membrane can capture COPI vesicles to direct them to the Golgi. Abbreviations: CDG, congenital disorder of glycosylation; COG complex, conserved oligomeric Golgi complex; COPI, coat protein complex I; SNARE, soluble N-ethylmaleimide-sensitive factor attachment protein receptor.

**Figure 4 ijms-21-04654-f004:**
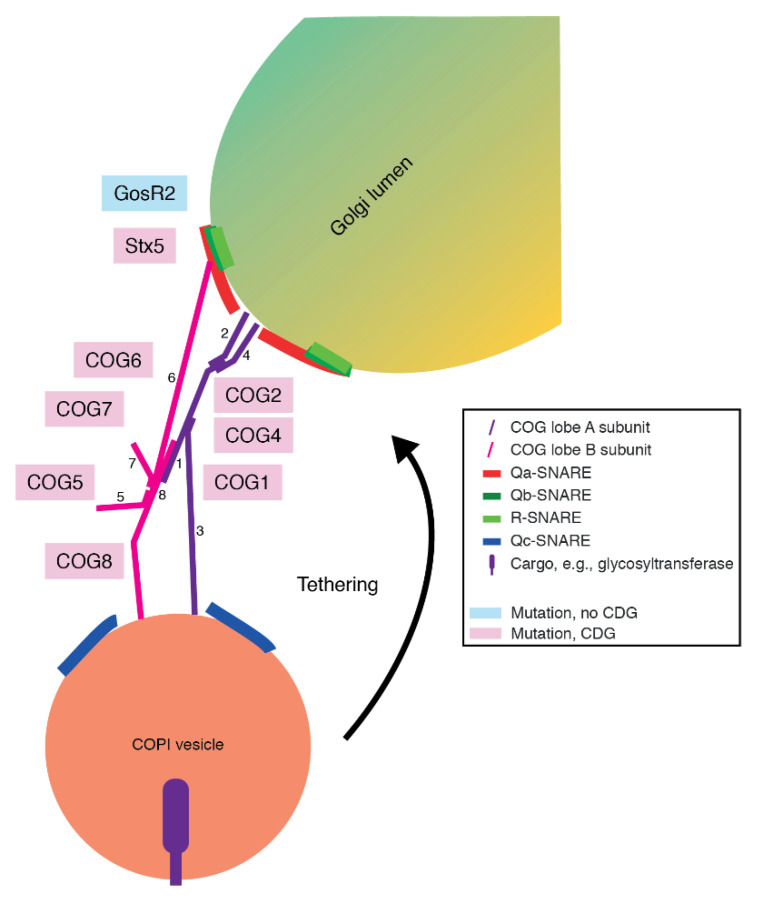
Schematic overview of COG-mediated vesicle tethering and SNARE-mediated vesicle fusion at the Golgi. COPI vesicles are tethered to the Golgi membrane through the interaction of COG lobes A and B. Subsequently, SNARE-mediated membrane fusion occurs, and the vesicle cargo is released to the Golgi lumen. Abbreviations: CDG, congenital disorder of glycosylation; COG complex, conserved oligomeric Golgi complex; COPI, coat protein I; SNARE, soluble N-ethylmaleimide-sensitive factor attachment protein receptor.

**Table 1 ijms-21-04654-t001:** Overview of membrane trafficking-related CDGs and their phenotypes.

Gene	Mutation	CDG ^a^	Clinical Phenotype ^b^	N-Glycosylation ^c^	Man	GlcNAc	Gal	Sia	O-Glycosylation ^c^	Screening ^d^	References
*ATP6V0A2*	V66fsX107	X	+	−−−	=	=	−	−−	−−	IEF, MS	[48]
*ATP6V0A2*	T643fsX683	X	+	−−−	=	=	−	−−	n.d.	IEF	[48]
*ATP6V0A2*	Q765X	X	+	−−−	=	=	−	−−	−−	IEF, MS	[48]
*ATP6V0A2*	R63X	X	++	−−−	=	=	−	−−	−−	IEF, MS	[48,225]
*ATP6V0A2*	K117fsX144	X	++	−−−	=	=	−	−−	−−	IEF, MS	[48,225]
*ATP6V0A2*	n.d.	X	++	−−−	=	=	−	−−	−−	IEF, MS	[48,225]
*ATP6V0A2*	D243fsX258 and E442fsX506	X	++	−−−	=	=	−	−−	=	IEF, MS	[48]
*ATP6V0A2*	T280fsX285	X	+++	−−−	=	=	−	−−	−−	IEF, MS	[48]
*ATP6V0A2*	E442X	X	+++	−−−	=	=	−	−−	=	IEF, MS	[48]
*ATP6AP1*	M4281	X	+	−−−	=	=	−−	−−	−−	IEF, MS	[92]
*ATP6AP1*	L144P	X	+	−−−	=	=	−−	−−	−−	IEF, MS	[92]
*ATP6AP1*	E346K	X	++	−−−	=	=	−−	−−	−−	IEF, MS	[92]
*ATP6AP1*	Y313C	X	++	−−−	=	=	−−	−−	−−	IEF, MS	[92]
*ATP6AP2*	L98S	X	++	−−−	=	=	−−	−−	=	CZE, IEF, MS	[93]
*ATP6AP2*	L98S	X	+	−−−	=	=	−−	−−	=	CZE, IEF, MS	[93]
*ATP6AP2*	R71H	X	++	−−−	=	=	−−	−−	=	CZE, IEF, MS	[93]
*VMA21*	n.d.^1^	X	*+*	*−−*	=	=	*−−*	*−−*	−	IEF, MS	[97]
*VMA21*	R18G*	X	*+*	*−−*	=	=	*−−*	*−−*	−	IEF, MS	[97]
*VMA21*	N63G	X	*+*	*−−*	=	=	*−−*	*−−*	−	IEF, MS	[97]
*TMEM199*	A7E	X	+	−−−	=	=	−	−	−−−	IEF, MS	[102]
*TMEM199*	A14P	X	+	−−−	=	=	−	−	=	IEF, MS	[102]
*TMEM199*	R31P	X	+	−−−	=	=	−	−	−−−	IEF, MS	[102]
*CCDC115*	L31S	X	++	−−−	=	=	−−	−−	−−−	IEF, MS	[101]
*CCDC115*	D11Y	X	++	−−−	=	=	−−	−−	−−−	IEF, MS	[101]
*TMEM165*	n.d.^2^	X	++	−−−	=	=	−−	−−	=	IEF, MS, L	[111]
*TMEM165*	R126C	X	++	−−−	=	=	−−	−−	=	IEF, MS, L	[111]
*TMEM165*	R126C and G304R	X	+	−−−	=	=	−−	−−		IEF, MS, L	[111]
*GMAP-210*	L1668X	−	+++	−−−	=	=	−−	−−	=	L	[117]
*GMAP-210*	G439VfsX20 and n.d.^4^	−	+++	n.d.	n.d.	n.d.	n.d.	n.d.	n.d.	Blot	[120]
*GMAP-210*	D410Y and E1606LfsX3	−	+++	n.d.	n.d.	n.d.	n.d.	n.d.	n.d.	Blot	[120]
*GMAP-210*	Q196X and Q1512X	−	+++	n.d.	n.d.	n.d.	n.d.	n.d.	n.d.	Blot	[120]
*GMAP-210*	D410Y and I710CfsX19	−	+++	n.d.	n.d.	n.d.	n.d.	n.d.	n.d.	Blot	[120]
*GMAP-210*	K541RfsX17 and M1806V	−	+++	n.d.	n.d.	n.d.	n.d.	n.d.	n.d.	Blot	[120]
*GMAP-210*	Q196X and K998SfsX5	−	+++	n.d.	n.d.	n.d.	n.d.	n.d.	n.d.	Blot	[120]
*GMAP-210*	Q196X and R264X	−	+++	n.d.	n.d.	n.d.	n.d.	n.d.	n.d.	Blot	[120]
*GOLGB1*	n.d.^3^	−	++	−−−	=	=	−−	−−	−−−	L	[121]
*GORAB*	F8L	−	++	−−−	=	=	=	−−	=	MS, L	[125]
*GORAB*	K190del	−	++	−−−	=	=	=	−−	=	MS, L	[125]
*GORAB*	M1?	X	++	=	=	=	=	=	=	CZE	[127,226]
*GORAB*	E46X	X	++	n.d.	=	=	=	=	n.d.	n.d.	[127]
*GORAB*	P86RfsX70	X	++	n.d.	=	=	=	=	n.d.	n.d.	[127]
*GORAB*	E123X	X	++	n.d.	=	=	=	=	n.d.	n.d.	[127]
*GORAB*	S175_R221del	X	++	n.d.	=	=	=	=	n.d.	n.d.	[127]
*GORAB*	Q247X	X	++	n.d.	=	=	=	=	n.d.	n.d.	[127]
*GORAB*	R262X	X	++	n.d.	=	=	=	=	n.d.	n.d.	[127]
*GORAB*	F350LfsX26	X	++	n.d.	=	=	=	=	n.d.	n.d.	[127]
*SCYL1*	H392PfsX30	−	+	n.d.	n.d.	n.d.	n.d.	n.d.	n.d.	n.d.	[133]
*SCYL1*	V313CfsX6 and n.d.^4^	−	+	n.d.	n.d.	n.d.	n.d.	n.d.	n.d.	n.d.	[134]
*SCYL1*	A504PfsX15 and Q546X	−	+	n.d.	n.d.	n.d.	n.d.	n.d.	n.d.	n.d.	[134]
*SCYL1*	Q57X	−	+	n.d.	n.d.	n.d.	n.d.	n.d.	n.d.	n.d.	[132]
*SCYL1*	E86X	−	+	n.d.	n.d.	n.d.	n.d.	n.d.	n.d.	n.d.	[132]
*SCYL1*	A105V	−	+	n.d.	n.d.	n.d.	n.d.	n.d.	n.d.	n.d.	[132]
*SCYL1*	V313CfsX6 and Q347X	−	+	n.d.	n.d.	n.d.	n.d.	n.d.	n.d.	n.d.	[132]
*SCYL1*	n.d.^4^	−	+	n.d.	n.d.	n.d.	n.d.	n.d.	n.d.	n.d.	[132]
*SCYL1*	D478G	−	+	n.d.	n.d.	n.d.	n.d.	n.d.	n.d.	n.d.	[132]
*SCYL1*	A504PfsX15	−	+	n.d.	n.d.	n.d.	n.d.	n.d.	n.d.	n.d.	[132]
*SCYL1*	Q546X	−	+	n.d.	n.d.	n.d.	n.d.	n.d.	n.d.	n.d.	[132]
*SCYL1*	Q628X	−	+	n.d.	n.d.	n.d.	n.d.	n.d.	n.d.	n.d.	[132]
*COG1*	900X	X	++	−−−	=	=	−−	−−	−−−	IEF, MS, L	[153]
*COG1*	n.d.^4^	X	++	−−	=	=	−	−−	−−	IEF, MS	[161]
*COG2*	Y234X and W634G	X	+++	−−	=	=	−	−−	n.d.	IEF, MS	[152]
*COG4*	L773R	X	++	−−−	=	=	=	−−	−−−	IEF, HPLC, MS	[162,163]
*COG4*	E233X	X	++	−−−	=	=	=	−−	−−−	IEF, HPLC, MS	[162,163]
*COG4*	R729W	X	++	−−−	=	=	=	−−	=	IEF, MS	[164]
*COG4*	G516R	X	++	=	=	=	=	=	=	MS	[165]
*COG5*	n.d.^4^	X	+/++	−−	+	+	−−	−−	−−	IEF, MS	[154,166,167,168]
*COG6*	G549V	X	+++	−−	=	=	−	−−	−−	IEF, HPLC, MS	[169,170,171,172]
*COG7*	n.d.^4^	X	+++	−−−	=	=	=	−−	−−−	IEF, L	[157,158,174,175]
*COG7*	n.d.^4^	X	+++	−−	=	=	−	−−	−−	IEF, MS	[173]
*COG8*	Y537X	X	++	−−−	=	=	−	−	−−−	IEF, MS, L	[160]
*COG8*	F563HfsX4	X	++	−−−	=	=	=	−	−−−	MS, L	[159]
*STX5*	M55V	X	+++	−−−	++	=	=	−−	−−−	IEF, MS, L	[191]
*SNAP29*	R29X	−	++	n.d.	=	=	=	=	n.d.	n.d.	[201]
*SNAP29*	L119AfsX15 and n.d.^4^	−	++	n.d.	=	=	=	=	n.d.	n.d.	[202]
*SNAP29*	R85X	−	++	n.d.	=	=	=	=	n.d.	n.d.	[203]
*SNAP29*	T130fsX17	−	++	n.d.	=	=	=	=	n.d.	n.d.	[204]
*SNAP29*	P10fsX42	−	++	n.d.	=	=	=	=	n.d.	n.d.	[204]
*SNAP29*	R90C	−	++	n.d.	=	=	=	=	n.d.	n.d.	[204]
*SNAP29*	E89K	−	++	n.d.	=	=	=	=	n.d.	n.d.	[204,205]
*SNAP29*	V75fsX28	−	+++	=	=	=	=	=	=	?	[206]
*SNAP29*	S163fsX5	−	++	n.d.	=	=	=	=	n.d.	n.d.	[207]
*GOSR2*	G144W	−	++	n.d.	n.d.	n.d.	n.d.	n.d.	n.d.	n.d.	[209,210]
*GOSR2*	K164del	−	+	n.d.	n.d.	n.d.	n.d.	n.d.	n.d.	n.d.	[209]

^a^ X, confirmed congenital disorder of glycosylation; -, unconfirmed congenital disorder of glycosylation. ^b^ Severity of clinical symptoms. +, mild symptoms; ++, moderate symptoms; +++, severe symptoms. ^c^ Severity of glycosylation phenotype. −−−, strong decrease; −−, moderate decrease; −, mild decrease; =, no change; +, mild increase; ++, moderate increase; +++, strong increase; n.d., not determined. ^d^ CZE, capillary zone electrophoresis; IEF, isoelectric focusing; HPLC, high-performance liquid chromatography; MS, mass spectrometry; L, lectin staining. ^1^ New ATG start codon upstream of first exon, causing a frameshift and premature stop codon at c.26-28. ^2^ Cryptic splice donor site activation. ^3^ Intronic splice donor site; causes S793F in *ITGB5*. ^4^ Intronic splice site.

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
