# Peer review of "Sugary Logistics Gone Wrong: Membrane Trafficking and Congenital Disorders of Glycosylation"

_ijms, 2020, doi:10.3390/ijms21134654_

Round 1

Reviewer 1 Report

This is a well written and interesting review on how membrane trafficking defects at the Golgi apparatus lead to diseases associated with defective glycosylation. It is timely and up to date, with a comprehensive citation of the relevant literature. I think it makes a valuable contribution to the field. There are some aspects that I feel need attention though. Once these have been addressed, I would be happy to recommend publication.

Major points:

  • The section on vacuolar ATPase mentions that changes in luminal pH most likely cause defective glycosylation by interfering with enzyme trafficking. However, another possibility is that the altered luminal pH directly affects enzyme activity by changing it from the pH optima for the enzymes. This point is discussed in the general discussion, but I felt this was too late on in the review. It should be mentioned near the beginning, ideally in the v-ATPase section. Also, although it is mentioned that a small change in luminal pH is unlikely to affect enzyme activity, can this possibility be excluded? If not, then it should be given credence as an additional plausible disease mechanism.
  • In the v-ATPase section what is lacking is any sense of where assembly of the v-ATPase occurs, and whether the trafficking of this complex itself is actually important. To what extent do the different mutations affect v-ATPase assembly versus its trafficking? The text and Fig 2 should be modified to clarify this aspect. The legend to Fig 2 could also more extensive to better describe the different v-ATPase subunits.
  • Line 206: The text mentions ER to Golgi transport being sensitive to loss of v-ATPase activity. But surely the major effect will be on intra-Golgi recycling of enzymes? In a few other places the text could be clearer whether it is intra-Golgi recycling of enzymes that is affected, or ER to Golgi transport of the cargo or the enzymes that is altered. I would suspect in most cases it would be the intra-Golgi trafficking. Please can the authors check this and clarify where appropriate.
  • The proteins GORAB and GRASPs are described in the Golgins and Rabs section. However, neither protein is a golgin, according to the definition of these proteins. In addition, although GRASPs do mediate membrane tethering events, and so could be thought to function in an analogous way to golgins, GORAB is not thought to, rather participating in vesicle assembly or formation. I think the information needs to be reorganized accordingly. There are also some issues with Fig 3. GORAB is drawn like a golgin and looks to be tethering a vesicle. It is not involved in this process, instead participating in vesicle formation. The scale of the proteins is wrong, GORAB is quite small compared to the golgins, and it does not bind SNAREs, as the figure suggests. The figure should be redrawn.
  • The referencing on GMAP-210 could be improved. There have been more recent papers than the Smits one describing patient mutations and defects in glycosylation. For example, it has been shown that complete LOF alleles cause ACG1A, whereas hypomorphic mutations cause a milder disease called ODCD, with intermediate effects on trafficking and glycosylation (Wehrle et al, 2019). The Table should also include these more recently identified mutations. It should also be made clearer that it is defects in glycosylation of extracellular matrix proteins that is causing the patient phenotype. It should also be mentioned that GMAP-210 is involved in tethering of both ER to Golgi and intra-Golgi vesicles.
  • In the section on GORAB, it should be mentioned that there is a deficit in proteoglycan glycosylation, with decorin and biglycan highly affected, and that the patient phenotype is due to a defect in matrix protein modification. Also, in this section it is mentioned GORAB might influence early and late Golgi trafficking. However, GORAB is only at the TGN, so this is not likely. SCYL1 is present earlier on, as the authors mention, but its mutation gives a very different phenotype (liver and brain mainly). If there are glycan changes involved, which would seem likely, it might be worth including Scyl1 in the Table, and describing it differently in the main text.

Minor points:

Line 53: text states Golgi exit; it should be entry

Line 58: text states cis-Golgi closest to centrosome. I’m not sure this is actually correct. Better I think would be to say that cis-Golgi is closest to the ER. Also, it needs to be made explicit here that the cis-Golgi is where ER-derived cargo enters the Golgi, and that it exits at the TGN.

Line 103: I would remove Arl1 from the list of proteins mentioned. Unlike the other proteins mentioned, it is not involved in intra-Golgi transport, so not really relevant for the topic of this review.

Fig 1: There is a label for fusion in the diagram, but it is not clear at all what it is meant to be showing. No fusion is depicted in the diagram.

Line 175: The text mentions ‘impaired lipophagy due to the loss of sialic acid and galactose on glycoproteins’. The mechanism here is unclear, and more explanation is required as to how defective Golgi glycosylation could cause this phenotype.

Related to the sections of golgins and COG: the authors may want to refer to the following review which describes how the golgins and COGs likely cooperate for vesicle tethering at the Golgi: Witkos and Lowe, Recognition and Tethering of Transport Vesicles at the Golgi Apparatus, Current Opinion in Cell Biology, 2017.

The paragraph on COG starting at line 310 is hard to follow. The authors need to explain better the significance of the membrane anchoring experiments. What is the conclusion from them?

Line 368: A concluding sentence is needed here to summarise the main points.

Table: Are all the known COG patient mutations mentioned? There don’t seem to be many.

Author Response

This is a well written and interesting review on how membrane trafficking defects at the Golgi apparatus lead to diseases associated with defective glycosylation. It is timely and up to date, with a comprehensive citation of the relevant literature. I think it makes a valuable contribution to the field. There are some aspects that I feel need attention though. Once these have been addressed, I would be happy to recommend publication.

Major points:

  • The section on vacuolar ATPase mentions that changes in luminal pH most likely cause defective glycosylation by interfering with enzyme trafficking. However, another possibility is that the altered luminal pH directly affects enzyme activity by changing it from the pH optima for the enzymes. This point is discussed in the general discussion, but I felt this was too late on in the review. It should be mentioned near the beginning, ideally in the v-ATPase section. Also, although it is mentioned that a small change in luminal pH is unlikely to affect enzyme activity, can this possibility be excluded? If not, then it should be given credence as an additional plausible disease mechanism.

We weakened our statement on page 2 and have added a short passage on the effect of abnormal luminal pH on glycosyltransferase activity at the end of the first paragraph of the V-ATPase section (page 4). Here we have also added that both the reduced glycosyltransferase activity and their mislocalization likely contribute to disease.

  • In the v-ATPase section what is lacking is any sense of where assembly of the v-ATPase occurs, and whether the trafficking of this complex itself is actually important. To what extent do the different mutations affect v-ATPase assembly versus its trafficking? The text and Fig 2 should be modified to clarify this aspect. The legend to Fig 2 could also more extensive to better describe the different v-ATPase subunits.

We now discuss in a bit more detail V-ATPase assembly in Saccharomyces cerevisiae and discuss the possibility that the assembly is similar in mammalian cells (page 6). Unfortunately, the exact mechanism of V-ATPase assembly in mammals is not as well understood as in yeast and we cannot give a definite answer to the assembly mechanism. Some evidence exists as TMEM199 and CCDC115 localize to the ER membrane (Miles et al., eLife 2017; cited in the manuscript) but more extensive characterization is necessary. We now discuss this in the text (page 6), and we have updated figure 2 to show the ER membrane and the assembly step involving TMEM199 and CCDC115. We have also amended the legend to figure 2 to clarify the V-ATPase subunits.

  • Line 206: The text mentions ER to Golgi transport being sensitive to loss of v-ATPase activity. But surely the major effect will be on intra-Golgi recycling of enzymes? In a few other places the text could be clearer whether it is intra-Golgi recycling of enzymes that is affected, or ER to Golgi transport of the cargo or the enzymes that is altered. I would suspect in most cases it would be the intra-Golgi trafficking. Please can the authors check this and clarify where appropriate.

We previously used ER-Golgi trafficking as an umbrella term for both ER-to-Golgi and intra-Golgi trafficking but now realize that this may be confusing. We thank the reviewer for pointing this out and have now updated the text in several places throughout the manuscript to use the more appropriate terminology of intra-Golgi trafficking.

  • The proteins GORAB and GRASPs are described in the Golgins and Rabs section. However, neither protein is a golgin, according to the definition of these proteins. In addition, although GRASPs do mediate membrane tethering events, and so could be thought to function in an analogous way to golgins, GORAB is not thought to, rather participating in vesicle assembly or formation. I think the information needs to be reorganized accordingly. There are also some issues with Fig 3. GORAB is drawn like a golgin and looks to be tethering a vesicle. It is not involved in this process, instead participating in vesicle formation. The scale of the proteins is wrong, GORAB is quite small compared to the golgins, and it does not bind SNAREs, as the figure suggests. The figure should be redrawn.

We have renamed section 2.2 “Golgins, GRASPs, GORAB and Rabs” to clearly separate GORAB and GRASPs from Golgins and to not specifically name GRASPs as Golgins. We have amended the text to also reflect this (pages 7-8). In addition, we have addressed the errors in figure 3 and now emphasize that GORAB binds COPI components.

  • The referencing on GMAP-210 could be improved. There have been more recent papers than the Smits one describing patient mutations and defects in glycosylation. For example, it has been shown that complete LOF alleles cause ACG1A, whereas hypomorphic mutations cause a milder disease called ODCD, with intermediate effects on trafficking and glycosylation (Wehrle et al, 2019). The Table should also include these more recently identified mutations. It should also be made clearer that it is defects in glycosylation of extracellular matrix proteins that is causing the patient phenotype. It should also be mentioned that GMAP-210 is involved in tethering of both ER to Golgi and intra-Golgi vesicles.

We now include a passage on ODCD referencing Wehrle et al., and we discuss the effects of the mutations on extracellular matrix proteins (page 7). In addition, we have added the more recently identified mutations to Table 1. We have also added a sentence on the function of GMAP-210 in tethering both ER-to-Golgi and intra-Golgi vesicles.

  • In the section on GORAB, it should be mentioned that there is a deficit in proteoglycan glycosylation, with decorin and biglycan highly affected, and that the patient phenotype is due to a defect in matrix protein modification. Also, in this section it is mentioned GORAB might influence early and late Golgi trafficking. However, GORAB is only at the TGN, so this is not likely. SCYL1 is present earlier on, as the authors mention, but its mutation gives a very different phenotype (liver and brain mainly). If there are glycan changes involved, which would seem likely, it might be worth including Scyl1 in the Table, and describing it differently in the main text.

We have added the proteoglycan glycosylation deficiencies as an important factor in the patient phenotypes (pages 7-8). Additionally, we have changed the wording to reflect that GORAB is only functional at late Golgi compartments. We have found publications describing in mutations in Scyl1 (Lenz, et al. (2018), Schmidt, et al. (2007), Schmidt, et al. (2015); all cited in the manuscript), one of which describes glycosylation abnormalities for one patient. We have added a short section on Scyl1 mutations (page 8) and added the mutations to Table 1.

Minor points:

Line 53: text states Golgi exit; it should be entry

We corrected the text as suggested by the reviewer.

Line 58: text states cis-Golgi closest to centrosome. I’m not sure this is actually correct. Better I think would be to say that cis-Golgi is closest to the ER. Also, it needs to be made explicit here that the cis-Golgi is where ER-derived cargo enters the Golgi, and that it exits at the TGN.

We have changed “closest to centrosome” to “closest to the ER” and we have added a small description of the passage of glycoproteins through the Golgi from cis-Golgi to TGN (page 2).

Line 103: I would remove Arl1 from the list of proteins mentioned. Unlike the other proteins mentioned, it is not involved in intra-Golgi transport, so not really relevant for the topic of this review.

We have removed Arl1 as requested by the reviewer.

Fig 1: There is a label for fusion in the diagram, but it is not clear at all what it is meant to be showing. No fusion is depicted in the diagram.

The fusion label has been removed from figure 1 to mitigate any confusion.

Line 175: The text mentions ‘impaired lipophagy due to the loss of sialic acid and galactose on glycoproteins’. The mechanism here is unclear, and more explanation is required as to how defective Golgi glycosylation could cause this phenotype.

We have clarified that the cause of the impaired lipophagy in these patients is deficient lysosomal acidification and that the defective Golgi glycosylation is a separate phenotype (page 6).

Related to the sections of golgins and COG: the authors may want to refer to the following review which describes how the golgins and COGs likely cooperate for vesicle tethering at the Golgi: Witkos and Lowe, Recognition and Tethering of Transport Vesicles at the Golgi Apparatus, Current Opinion in Cell Biology, 2017.

We now refer to this review in the introduction of the COG section.

The paragraph on COG starting at line 310 is hard to follow. The authors need to explain better the significance of the membrane anchoring experiments. What is the conclusion from them?

We have added a conclusion to section 2.3 by stating that the membrane association of COG subunits is necessary for their function. We have also added a sentence for clarity that COG subunits are membrane-associated proteins that lack a transmembrane domain (page 9-12)

Line 368: A concluding sentence is needed here to summarise the main points.

We have added a general conclusion to the COG complex section on how the described mutations reveal an indispensable role for the entire COG complex in intra-Golgi vesicle tethering and glycosylation (page 12).

Table: Are all the known COG patient mutations mentioned? There don’t seem to be many.

We were indeed missing several COG patient mutations. For COG1-CDG, we had described one mutation in the text but it had been omitted from the table. Moreover, we have added mutations and descriptions to the Table and the text for COG2, COG4, COG5 and COG6.

Reviewer 2 Report

In this review the authors bring together available data on protein glycosylation and how defects in this process lead to congenital disorders of glycosylation (CDG). The known genetic variations in proteins involved in pH maintenance and vesicular transport in the Golgi are presented and the effects on protein glycosylation and the patients symptoms discussed. Golgins which are not (yet) associated to CDGs but lead to an abnormal phenotype in mice or cells are included, making the story complete.

The review brings together information from different fields (cell biology, medical research, yeast biology) and is pleasant to read, the different parts coming in a logical, intuitive order. The figures are clear and colourful, as well as inserted in the right spots to help visualisation of the protein complexes involved.

Author Response

In this review the authors bring together available data on protein glycosylation and how defects in this process lead to congenital disorders of glycosylation (CDG). Thereby, they include the V-ATPase, responsible for the maintenance of the pH in the different Golgi compartments, but also proteins performing and controlling the vesicular transport from ER to Golgi and more specifically the recruitment of COP1 vesicles (the golgins and their Rab proteins, the conserved oligomeric Golgi complex (COG) and SNARES).

For all these proteins, the authors present the known genetic variations and discuss the effect on protein glycosylation as well as potential symptoms in patients. The story is completed by including Golgins which are not associated to CDGs but lead to an abnormal phenotype in mice (giantin) or cells in culture (giantin, GRASP55 and GRAPS65).

Finally, the yeast homologs to some Human proteins described are included, which gives interesting information about the function of the proteins involved, bringing together in the same review biological and medical aspects.

This review brings together cell biology and medical aspects. The order in which the topics are addressed is logical to the reader and the figures clear and colourful as well as inserted in the right spot to exemplify the text description, helping visualising the protein complexes and cellular processes involved.

Minor points:

53 shouldn’t it be ER exit instead of Golgi exit? Proteins haven’t reached the Golgi them, do they?

This was indeed incorrect; we have corrected the text to “Golgi entry”.

192 please change the order of the sentence “Abnormal glycosylation is seen for both CDGs, mainly the truncation of glycans through the loss of sialylation and galactosylation” into “Abnormal glycosylation, mainly the truncation of glycans through the loss of sialylation and galactosylation, is seen for both CDGs.”

We have changed the order of this sentence as per the reviewer’s suggestion.

363 in “Intronic mutations affecting the mRNA splicing and ultimately cause the decrease of protein levels of COG7 cause similar glycosylation defects” - the first “cause” needs to be changed into “causing”.

We have corrected this mistake.

390 “Contrary to plasma membrane SNARE localized complexes...” should be “Contrary to plasma membrane localized SNARE complexes” (SNARE and localized are exchanged).

We have exchanged the position of “SNARE” and “localized”.